# FIXING THAT FREE LUNCH: WHEN, WHERE, AND WHY SYNTHETIC DATA FAILS IN MODEL-BASED POLICY OPTIMIZATION

## ABSTRACT

Synthetic data is a core component of data-efficient Dyna-style model-based reinforcement learning, yet it can also degrade performance. We study when it helps, where it fails, and why, and we show that addressing the resulting failure modes enables policy improvement that was previously unattainable. We focus on Model-Based Policy Optimization (MBPO), which performs actor and critic updates using synthetic action counterfactuals. Despite reports of strong and generalizable sample-efficiency gains in OpenAI Gym, recent work shows that MBPO often underperforms its model-free counterpart, Soft Actor-Critic (SAC), in the DeepMind Control Suite (DMC). Although both suites involve continuous control with proprioceptive robots, this shift leads to sharp performance losses across seven challenging DMC tasks, with MBPO failing in cases where claims of generalization from Gym would imply success. This reveals how environment-specific assumptions can become implicitly encoded into algorithm design when evaluation is limited. We identify two coupled issues behind these failures: scale mismatches between dynamics and reward models that induce critic underestimation and hinder policy improvement during model-policy coevolution, and a poor choice of target representation that inflates model variance and produces error-prone counterfactuals in MBPO rollouts. Addressing these failure modes enables policy improvement where none was previously possible, allowing MBPO to outperform SAC in five of seven tasks while preserving the strong performance previously reported in OpenAI Gym. Rather than aiming only for incremental average gains, we hope our findings motivate the community to develop taxonomies that tie MDP task- and environment-level structure to algorithmic failure modes, pursue unified solutions where possible, and clarify how benchmark choices ultimately shape the conditions under which algorithms generalize.

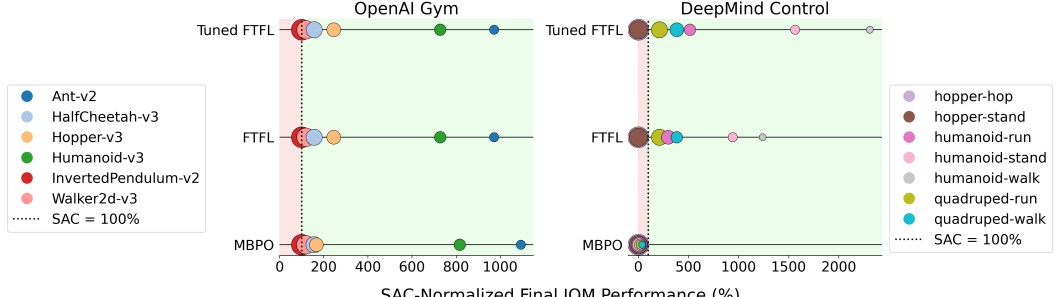

Figure 1: Per-environment interquartile mean (IQM) returns normalized to SAC in DeepMind Control Suite (DMC) and OpenAI Gym across 6 seeds per task. SAC is the baseline (100%), with values above shaded green (improvement) and below shaded red (underperformance). MBPO matches or exceeds SAC only in Gym, failing on all DMC tasks. Fixing That Free Lunch (FTFL) and Tuned FTFL outperform SAC on all DMC tasks except `hopper-hop` and `hopper-stand`, with large gains in most cases. Task circle sizes are reduced left to right for visibility of overlaps.

# 1 INTRODUCTION

The "no free lunch" principle highlights that no optimization method achieves universal optimality across every problem instance. In reinforcement learning (RL), this underscores that an algorithm's effectiveness is shaped by the specific environment and its characteristics. For Dyna style model-based RL (MBRL) algorithms, wherein synthetic model-generated data is intended to improve sample efficiency and enable rapid policy improvement, recent evidence suggests that these benefits are not guaranteed. In *Stealing That Free Lunch* (STFL) (Barkley & Fridovich-Keil, 2025), Model Based Policy Optimization (MBPO) (Janner et al., 2019), a widely used and well cited Dyna style algorithm, was shown to match or exceed its model free counterpart, Soft Actor Critic (SAC) (Haarnoja et al., 2019), in OpenAI Gym (Brockman et al., 2016). However, in many DeepMind Control Suite (DMC) (Tassa et al., 2020) tasks, MBPO then failed to improve beyond a randomly initialized policy when trained from scratch, despite both benchmarks sharing the same MuJoCo physics engine (Todorov et al., 2012) and featuring similar high level task reward structures.

While STFL established that these failures exist, it did not explain why MBPO collapses in specific settings, other than noting the central role of synthetic data in driving failures. In this work, we close that gap. We focus on challenging DMC tasks where MBPO failed completely in STFL, often performing no better than a random initial policy throughout training (Barkley & Fridovich-Keil, 2025). Through targeted experiments, we identify two coupled mechanisms driving these failures.

First, scale mismatches between dynamics and rewards in DMC cause errors in the reward model which then induce persistent critic underestimation during model–policy coevolution. This disrupts the explore–exploit balance that SAC normally provides and limits the diversity of real experience, further stalling policy improvement. Second, although residual prediction is common practice in MBRL (Nagabandi et al., 2018; Lambert et al., 2022), in the failing DMC tasks it inflates variance in the learned dynamics. This heightened uncertainty makes synthetic action counterfactuals unreliable despite seemingly low one-step model error, destabilizing MBPO's training and preventing effective policy improvement.

We find that with two minimal interventions we can break this loop: (1) applying target normalization independently to next state and reward predictions and (2) predicting next-state directly rather than residuals. With the success of these two interventions, we propose a new approach called Fixing That Free Lunch (FTFL). Through ablations we show that neither intervention alone recovers performance, but together they allow MBPO to improve in settings where it previously failed. As shown in Figure 1, FTFL preserves MBPO's strong Gym performance and makes learning possible in DMC, surpassing SAC in five of seven tasks with identical hyperparameters. Additionally, allowing for expanded model capacity (Tuned FTFL) further improves results, highlighting strong scaling potential.

Overall our contributions are:

1. **Diagnosis.** We empirically identify two coupled mechanisms behind MBPO's collapses on DMC: (a) output-target scale mismatch (next-state vs. reward) that suppresses reward learning and induces critic underestimation and (b) variance inflation from residual targets that increases synthetic transition error and destabilizes MBPO's policy improvements.

2. **FTFL:** We introduce two simple, lightweight modifications to MBPO that enable large gains in DMC where performance was previously no better than a random policy.

3. **Case study in algorithm–environment coevolution:** Narrow benchmark task sets can encode implicit assumptions into algorithm design and mask weaknesses that lead to poor generalization. We show that MBPO exemplifies this dynamic, over-promising in Gym but failing completely in seven of DMC's fifteen challenging continuous control robotics tasks. FTFL provides evidence that such weaknesses can be systematically addressed without sacrificing performance in Gym.

More broadly, our results provide evidence that although improving average performance across larger benchmark sets is necessary for building trust, averages can hide critical flaws. After all, RL practitioners expect algorithms to work, with modest tuning, on specific problems, and average gains are of little value if we cannot explain or anticipate failures in particular tasks. TD3 (Fujimoto et al., 2018) provides a convenient example: it is known to perform well with dense rewards, fail

in sparse settings, and succeed under sparse rewards when combined with Hindsight Experience Replay (Andrychowicz et al., 2018). This reflects an understanding of how structural properties of the underlying MDP (reward sparsity) shape algorithmic success. In much the same way, our findings underscore this need by providing a similar mapping: interactions between dynamics and reward scales, together with model training choices, can determine whether MBPO succeeds or fails even across seemingly similar tasks. Alongside higher average performance across diverse benchmarks, building up a body of prescriptive task–algorithm mappings is a necessary step toward ensuring RL methods are both effective and deployable in practice.

## 2 BACKGROUND

### 2.1 REINFORCEMENT LEARNING, MODEL-BASED RL, AND DYNA-STYLE ALGORITHMS

Reinforcement learning (RL) formalizes the problem of sequential decision making as a Markov Decision Process (MDP), defined by the tuple $(\mathcal{S}, \mathcal{A}, p, r, \gamma)$. Here, $\mathcal{S}$ is the state space, $\mathcal{A}$ is the action space, the probability distribution $p(s' \mid s, a)$ represents the transition dynamics, $r(s, a)$ is the reward function, and $\gamma \in [0, 1)$ is the discount factor. The goal of the agent is to learn a policy $\pi(a \mid s)$ that maximizes expected discounted reward.

Learning effective policies often requires extensive interaction with the environment, which is costly in both time and computation. Model-based RL methods address this challenge by learning a model of the environment's dynamics, $p_\theta(s', r \mid s, a)$, which can then be used to generate additional training data without further environment interaction. Within this paradigm, the Dyna architecture (Sutton, 1991) plays a central role: a learned world model $p_\theta$ is used to simulate hypothetical transitions, which are combined with real data to accelerate learning of an actor (a neural network policy) and a critic (a neural network value function).

A widely studied example is Model-Based Policy Optimization (MBPO) (Janner et al., 2019), which trains an ensemble of probabilistic neural networks to model state transitions and rewards. MBPO augments a replay buffer of real transitions with short model-generated rollouts branching from stored states. These synthetic transitions are then mixed with real data to update both the actor and the critic. MBPO builds on Soft Actor-Critic (SAC) (Haarnoja et al., 2018), a model-free algorithm noted for its stability and strong performance in continuous control tasks.

On the OpenAI Gym benchmark suite, MBPO consistently demonstrates strong performance and high sample efficiency, often outperforming its model-free counterparts. This success has made MBPO the foundation for a wide range of subsequent work (Lai et al., 2020; 2022; Zheng et al., 2023; Wang et al., 2023; Li et al., 2024; Dong et al., 2024).

## 3 RELATED WORK

### 3.1 GENERALIZATION ISSUES IN DYNA-STYLE REINFORCEMENT LEARNING

Benchmark-driven evaluations are the de facto standard in reinforcement learning, yet their validity as tests of generalization are rarely questioned. *Can We Hop in General?* (Voelcker et al., 2024) shows that benchmark choice can drastically alter conclusions, highlighting with Hopper environments that even different implementations of the same intuitive task yield contradictory results. Algorithms such as SAC (Haarnoja et al., 2019), MBPO (Janner et al., 2019), ALM (Ghugare et al., 2023), and DIAYN (Eysenbach et al., 2019) all report successes on some Hopper variants but fail on others, despite ostensibly targeting the same control problem. This reveals a broader issue: RL benchmarks are often treated as proxies for general capability, yet they may not even be representative of each other.

Building on this critique, *Stealing That Free Lunch* (STFL) (Barkley & Fridovich-Keil, 2025) systematically studied two Dyna-style methods, MBPO and ALM, across six OpenAI Gym and fifteen DeepMind Control Suite (DMC) tasks. Although all of these Gym and DMC tasks use MuJoCo (Todorov et al., 2012) and feature similar continuous control problems, MBPO and ALM's results diverged sharply: they consistently outperformed their base algorithms in Gym, yet failed in seven of fifteen DMC environments, often failing to outperform even a random initial policy. In the case

of MBPO, SAC not only outperformed MBPO across most DMC tasks, it did so with substantially less wall-clock time and compute. STFL investigated possible explanations — including overestimation bias, network plasticity, model fidelity, and extensive hyperparameter tuning — but found that none resolved the discrepancy. Crucially, it showed that introducing any synthetic rollout data into training consistently degraded policy performance.

Together, *Can We Hop in General?* (Voelcker et al., 2024) and *Stealing That Free Lunch* (Barkley & Fridovich-Keil, 2025) reveal how strongly RL results depend on benchmark choice and show that existing remedies fail to close the gap. Motivated by these findings, we turn to a focused analysis of MBPO to ask where, when, and why synthetic data causes its failures.

## 4 WHEN, WHERE, AND WHY SYNTHETIC DATA FAILS IN MBPO

STFL conducted a broad analysis of MBPO and found two critical results. First, remedies aimed solely at SAC's backbone, such as actor–critic resets, fail to resolve MBPO's underperformance in DMC. Second, introducing any amount of synthetic data into SAC's training pipeline causes immediate collapse in these environments. Together, these findings rule out deficiencies in SAC itself as the primary cause, and instead implicate the Dyna-style additions that distinguish MBPO. Building on this evidence, we narrow our focus in this paper to MBPO's model–based components and analyze the mechanisms by which synthetic data undermines policy improvement.

With this focus established, we investigate the mechanisms by which synthetic data causes MBPO to fail in the DeepMind Control Suite (DMC) but not in OpenAI Gym. Through two case studies analyzing training dynamics, we isolate the root causes of this discrepancy and identify simple remediations that correct MBPO's performance in a representative DMC environment. These insights motivate the generalizable remediations proposed in Section 5, where we validate their effectiveness across a wide range of DMC and Gym environments.

To enable this analysis, we build directly on STFL's efficient reimplementation of MBPO, which replicated Gym results while simplifying rollouts to a one-step regime (Barkley & Fridovich-Keil, 2025). This implementation eliminates the potential confounder of accumulating model errors in longer-horizon rollouts, allowing us to attribute performance gaps in DMC to other factors.

### 4.1 EXPERIMENTAL SETUP

Our goal in this section is to probe MBPO's ensemble training in isolation and identify anomalous behavior that explains its collapse in DMC. The setup is designed to make ensemble training in settings where synthetic data is beneficial directly comparable to settings where it causes failure, and to serve as a controlled diagnostic that informs the mechanistic analysis that follows.

To this end, we selected one representative environment from Gym (`Walker2d-v3`) and one from DMC (`humanoid-stand`). Using SAC with the original STFL hyperparameters, we trained three agents per task to generate replay buffers. We then selected the buffer from the agent with the most consistently high returns, using it as a diagnostic tool to probe MBPO's model training. To mimic online data collection without the confounding effects of actor–critic collapse in MBPO, we gradually revealed transitions from this buffer to MBPO's dynamics ensemble. This allowed us to study model training under controlled conditions where degenerate replay buffers would not arise.

Within this setup, we logged the progression of numerous training metrics for both MBPO and its SAC backbone in search of anomalous behavior. The following subsections report our findings and identify the specific causes of MBPO's degeneracy in DMC.

### 4.2 INAPPROPRIATE MODEL TARGET SCALE LEADS TO SEVERE CRITIC UNDERESTIMATION

STFL reported that in many DMC tasks where MBPO failed to improve beyond a random policy, the critic produced pathological estimates: either massive underestimates on the order of $-10^8$ or predictions collapsing to near-zero return. To stabilize training, STFL introduced layer normalization (Ba et al., 2016) between critic layers, following prior work on bounding critic outputs (Ball et al., 2023; Nauman et al., 2024). While this reduced the most extreme pathologies, MBPO still failed to outperform SAC, with critics frequently converging to near-zero return.

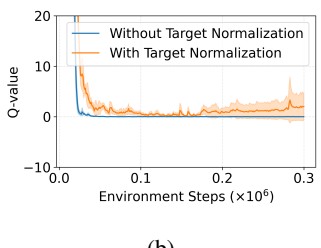
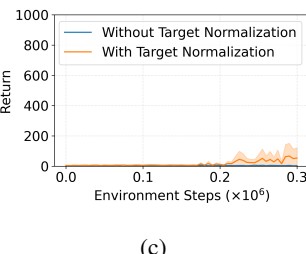

(a)          (b)          (c)

Figure 2: Poor scaling between reward and next-state predictions in MBPO's model leads to underestimated rewards, Q-value underestimation, and failure to improve beyond a random policy in `humanoid-stand`. (a) Real vs synthetic rewards with and without proper model target scaling, using the best performing SAC replay buffer (1 seed). (b) Critic estimates when MBPO is deployed online, 3 seeds (mean ± std). (c) Returns when MBPO is deployed online, 3 seeds (mean ± std).

This suggested that the critic was not the sole source of error. Using our pseudo-online training setup (Section 4.1) on `humanoid-stand`, we examined the quality of synthetic rewards generated by MBPO's one-step rollouts. Despite the replay buffers containing positive rewards much greater than 1 (DMC rewards are bounded between 0 and 1), MBPO's reward model consistently underestimated them, with predictions clustering near zero. The effect is clearest per seed, so we show one representative seed in Figure 2a, though the same underestimation appeared across multiple seeds.

Two implications follow. First, MBPO's joint regression target (state residuals alongside reward) prevents reward errors from being minimized effectively, leaving the reward model biased. Second, because synthetic transitions dominate critic training batch composition, these near-zero or negative rewards overwhelm the few accurate real transitions. This drives critic updates toward negative returns, explaining the severe underestimation observed in STFL and in our own online experiments.

We traced this failure to the training objective. MBPO trains its model on a multi-component target vector (next state and reward), where the relative scale of each component determines how well it can be learned (Goodfellow et al., 2016; LeCun et al., 1998). Because the model jointly regresses on both next states and rewards, any imbalance in their relative scales can suppress the reward signal. In practice, we observed that this mismatch led to systematic reward underestimation. While MBPO applies running unit normalization to model inputs, no analogous normalization is used for outputs. We hypothesize that this discrepancy is the direct cause of the reward model collapse.

To test this hypothesis, we introduced running unit normalization of each element of the target vector, dynamically rescaling next states and rewards using statistics from the replay buffer. By equalizing their contributions to the regression loss, this remediation led to results that validated our hypothesis. As shown in Figure 2a, normalization eliminated the reward model's collapse, driving prediction error to near zero. When deployed online, it prevented critic underestimation, allowing Q-values to rise with exploration (Figure 2b) and yielding consistent policy improvement on a DMC task where MBPO had previously failed (Figure 2c). This establishes the causal chain from improper target scaling to reward model bias, critic underestimation, and ultimately, policy failure.

**Key Insights:**
- In `humanoid-stand`, an output scale mismatch suppresses reward learning, collapsing synthetic rewards toward zero.
- Since synthetic transitions dominate critic training, this collapse drives severe value underestimation and prevents policy improvement.
- Applying output normalization remediates this failure, restoring reward learning and enabling policy improvement in `humanoid-stand`.
- Gym benchmarks masked this oversight in MBPO and many of its extensions, showing how benchmark choice can conceal algorithmic flaws.

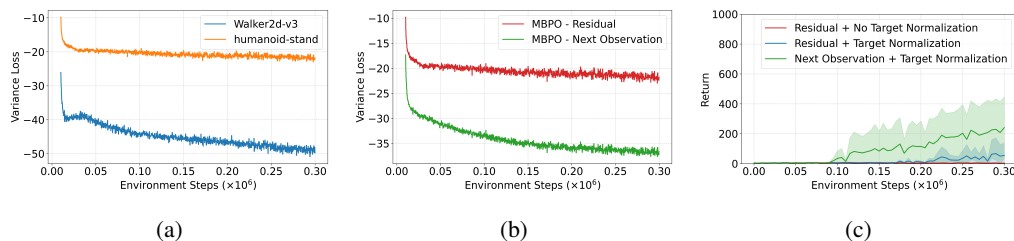

(a)  (b)  (c)

Figure 3: (a) Variance loss comparison between `walker2d` and `humanoid-stand`, showing substantially higher variance in the latter (1 seed). (b) Predicted variance under residual versus direct next-state modeling, where direct prediction yields lower variance and a more stable fit (1 seed). (c) Online returns for three ablation settings on `humanoid-stand`, demonstrating that only combining target normalization with direct prediction enables consistent policy improvement (3 seeds, mean ± std).

### 4.3 REVISITING RESIDUAL PREDICTION: A SUBOPTIMAL TARGET FOR MBPO

While output normalization enables MBPO to achieve its first better-than-random performance on `humanoid-stand`, the policy still underperforms its model-free SAC counterpart. This gap motivated us to search for systemic issues beyond scaling, beginning with the model's training objective. MBPO trains its ensemble with a negative log-likelihood objective, a form of heteroskedastic regression where each network predicts a mean $\mu$ and variance $\sigma^2$. Although DMC environments are deterministic, this uncertainty representation remains essential (Abbas et al., 2020): because the dynamics model is trained on a finite dataset, regions with limited or conflicting coverage appear stochastic. Predicting a variance allows the model to capture this data-driven uncertainty, signaling which predictions are less reliable. We refer to the component of the heteroskedastic negative log-likelihood that minimizes the predicted variance as the variance loss, which we use as a diagnostic signal in the following analysis.

To probe for instability, we compared the model's predicted variance on a successful task (`Walker2d-v3`) and a failing one (`humanoid-stand`), while utilizing target normalization as established in the previous section and operating in the pseudo-online manner described in Section 4.1. For each transition, we averaged ensemble-predicted variances $\sigma^2$ across features, networks, and training batches, yielding a metric directly comparable across state dimensionalities. As shown in Figure 3a, as depicted for one representative seed for clarity, variance in `humanoid-stand` is substantially higher than in `Walker2d-v3`, a clear diagnostic signal of a mismatch between the model's architecture and its prediction target.

This anomaly led us to revisit the standard practice of residual prediction, where models learn $s' - s$ rather than $s'$ directly. Residual prediction was popularized in early model-based RL approaches (Deisenroth & Rasmussen, 2011; Chua et al., 2018) and remains the default in modern world models (Hafner et al., 2021; 2023). The rationale is compelling: most state variables change slowly, so differences provide a strong inductive bias, allowing the model to focus on sparse dynamics rather than relearn the entire state at each step. This bias is particularly effective in smooth systems or for image-based states where consecutive frames are highly correlated (Sekar et al., 2020).

However, residual prediction is not universally superior. Its effectiveness depends strongly on system dynamics and noise characteristics (Lambert et al., 2018). In unstable regimes, residuals can amplify small errors that compound exponentially, producing rapid divergence. These results motivate the hypothesis that in DMC tasks, residual prediction inflates variance in the learned dynamics, yielding highly uncertain synthetic data that destabilizes MBPO's training.

To test this hypothesis, we replaced residual prediction with direct next-state prediction, combined with output normalization. As shown in Figure 3b for a representative seed, this change drastically reduced ensemble variance, indicating a better fit to the dynamics and a more stable, confident representation. Importantly, this improved representation translated into consistent policy improvement when MBPO was deployed online on `humanoid-stand`, as shown in Figure 3c, beyond even what was possible with just target normalization as shown in Section 4.2.

> **Key Insights:**
> - Residual prediction inflates variance in `humanoid-stand`, producing uncertain synthetic data that destabilizes MBPO.
> - Switching to next-state prediction with output normalization reduces variance and yields a more reliable representation.
> - This remediation enables stronger policy improvement than output normalization alone when MBPO is deployed online on `humanoid-stand`.
> - This provides further evidence that Gym benchmarks masked critical flaws, with DMC revealing algorithmic weaknesses hidden by seemingly similar tasks.

## 5 FIXING THAT FREE LUNCH: ELIMINATING (MOST) SYNTHETIC DATA ISSUES

Building on the coupled failures identified in Section 4, we propose two minimal modifications to MBPO that together restore generalization. We refer to this combined approach as *Fixing That Free Lunch (FTFL)*:

1. *Target Unit Normalization:* Apply running mean–variance normalization separately to next state and reward targets, balancing their relative scales in the model loss and preventing reward collapse.

2. *Direct Next state Prediction:* Replace residual prediction with absolute next state prediction. In DMC tasks this reduces inflated variance and yields a more reliable representation.

### 5.1 ABLATIONS: DO WE REALLY NEED BOTH?

To test whether the remedies identified in Sections 4.2 and 4.3 generalize beyond a single case, we performed ablations across the seven DMC tasks that MBPO previously failed in, as shown in Appendix C. FTFL succeeds in five of seven cases, enabling stable policy improvement where MBPO had previously failed. The two remaining environments, where both MBPO and STFL fail, are discussed in detail in Section 5.3. These results confirm that the failures identified earlier were not isolated artifacts but systemic issues that FTFL resolves in the majority of cases.

The ablations compared residual versus direct next state prediction, each trained with or without target normalization when FTFL is deployed online. Residual prediction without normalization reproduces the failures reported in STFL, with returns stuck near random performance. Adding normalization eliminates reward collapse and yields modest improvements across all environments, but returns remain low. Switching to direct next state prediction without normalization does little better, with performance still close to random, consistent with our results on `humanoid-stand` in Section 4.3. Only the combination of direct prediction with normalized targets, which defines FTFL, produces strong and stable returns across five of seven environments.

These results clarify how the two remedies interact. Target normalization restores balance between reward and dynamics learning, while direct prediction reduces model variance. Together, they resolve MBPO's coupled failures in most DMC tasks.

### 5.2 GENERALIZATION AND RESTORING THE PROMISE OF MBPO

Having established that FTFL resolves the coupled failures identified in Section 4, we evaluate it on the seven challenging DMC tasks where MBPO previously showed no improvement over a random policy. As shown in Figures 1 and 4, the gains are substantial. FTFL enables significant policy improvement in five of the seven tasks (`humanoid-stand`, `humanoid-walk`, `humanoid-run`, `quadruped-walk`, `quadruped-run`), consistently surpassing SAC and delivering reliable learning where MBPO collapsed. Furthermore, increasing the capacity of the model ensemble allows FTFL to surpass SAC by an even larger margin in the `quadruped` tasks, indicating the scalability of our approach.

These results are significant because they partially restore the original promise of MBPO. Leveraging a learned model for synthetic data can yield better final performance and improved sample efficiency. Where MBPO previously failed completely, FTFL delivers on this promise in five DMC tasks, showing that the underlying idea is sound once key implementation details tied to environment properties are addressed.

However, the broader promise of MBPO was to serve as a generalizable algorithm. Demonstrating success only on DMC would risk repeating the earlier mistake of assuming generality from a single benchmark suite. To test this, we evaluated FTFL on the set of OpenAI Gym tasks where MBPO has historically performed well. As shown in Figure 5, FTFL matches MBPO's strong performance on Gym across all six tasks. This demonstrates that our modifications are not a DMC-specific adjustment, but a robust improvement that generalizes across different benchmark suites.

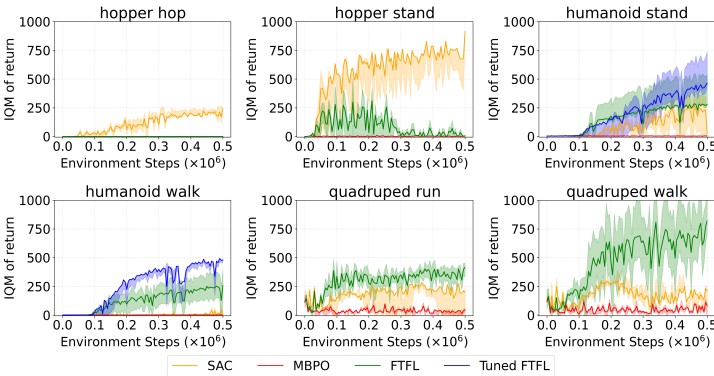

Figure 4: Return performance on six DMC tasks. Following Agarwal et al. (2022), solid lines show interquartile mean (IQM) returns aggregated across six seeds, and shaded regions denote 95% bootstrapped confidence intervals. Results are shown for FTFL with the original model and for Tuned FTFL with increased model capacity (effective only on `humanoid` tasks). One result was omitted for brevity, but raw averaged returns are provided for all seven tasks in Appendix B.

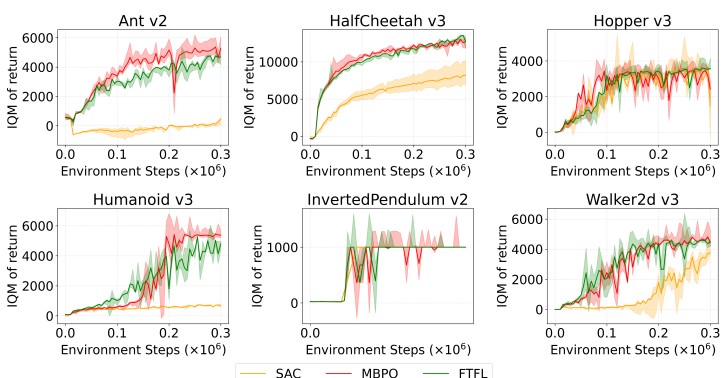

Figure 5: Return performance on six OpenAI Gym tasks, using the same IQM and confidence interval conventions as Figure 4. Raw averaged returns are provided in Appendix B.

### 5.3 REMAINING FAILURES AND TOWARDS A TAXONOMY OF RL FAILURE MODES

While FTFL resolves the failure modes in quadruped and humanoid environments, it does not fully mitigate MBPO's issues on `hopper-hop` and `hopper-stand`. In these Hopper tasks, scale and residual issues are corrected, reward collapse is prevented, and variance is reduced, consistent with the results in Section 5.1. Yet performance still stalls. An additional experiment (cf. Appendix D)

shows that removing the contact sensor in the `hopper` observation (a discontinuous contact-force signal) allows FTFL to match SAC on `hopper-stand`, but not on `hopper-hop`, pointing to a modality-specific challenge unique to Hopper that was absent in other failing DMC environments.

This discontinuous channel is poorly captured by the unimodal Gaussian next-state head with diagonal covariance. Unlike the remediations in FTFL, which generalize broadly, addressing this mismatch may require richer models that are not as transferable or require a-priori information.

This outcome should not be viewed as a regression, but as necessary motivation for future work. MBPO failed broadly across many DMC tasks. FTFL restores its effectiveness in quadruped and humanoid domains while unmasking the next layer of difficulty in Hopper. Once the dominant pathologies are repaired, subtler mismatches tied to discontinuous state channels become visible. Taken together with MBPO's original failings, these results underscore the need for a principled taxonomy of RL failure modes, organized along dimensions such as state smoothness and data modality. Such a framework would clarify how environment characteristics interact with algorithmic assumptions, and we return to its broader implications in the conclusion.

## 6 BROADER IMPLICATIONS AND CONCLUSION

Our findings reinforce the broader lesson that progress in reinforcement learning cannot be measured solely by higher average scores across increasingly large benchmark suites. Such averages mask the fact that algorithms often succeed in some environments while collapsing in others, and that these collapses are rarely random. Instead they emerge from structured interactions between environment characteristics and algorithmic design choices. Without a principled way to organize and explain these failures, researchers risk repeatedly rediscovering the same pitfalls under different names, and practitioners are left without guidance about when a method can be trusted. What is needed is a taxonomy of reinforcement learning failure modes, organized around dimensions such as state smoothness, reward scale, and model capacity. This taxonomy should systematically map environment properties to algorithmic assumptions, as is already done in practice for distinctions such as goal conditioned settings or sparse versus dense rewards.

Recent evidence strengthens this case. Palenicek et al. (2025) show that the CrossQ algorithm (Bhatt et al., 2024) is brittle in DeepMind Control and does not scale with increasing compute, but that weight normalization provides a simple and powerful stabilization that enables reliable scaling. Fujimoto et al. (2025) similarly highlight the no free lunch dynamic: MR.Q is the strongest method overall across continuous control benchmarks, yet it loses ground to TD7 (Fujimoto et al., 2023) in Gym and is surpassed by DreamerV3 (Hafner et al., 2023) in Atari only when DreamerV3 uses a model forty times larger, which then struggles to generalize elsewhere. These cases, like our own analysis of MBPO, illustrate that algorithmic success or failure depends critically on structural properties of the environment and implicit assumptions in the method.

Taken together, these results underscore the necessity of not only improving aggregate performance metrics, but also developing prescriptive mappings that explain where and why algorithms fail or underperform. Our work provides one such mapping in the form of a case study on MBPO. We identified two coupled mechanisms, mismatches between reward and dynamics scales and residual prediction variance inflation, that explain much of MBPO collapse in DeepMind Control. We then introduced two lightweight fixes, target normalization and direct next state prediction, that together resolve these failures and restore strong performance across most challenging tasks. FTFL, Fixing That Free Lunch, thus serves both as a practical solution for reliable use of synthetic data in MBPO and as an illustrative case study of how algorithm–environment mismatches can be diagnosed and repaired. In this way, FTFL provides not only a concrete solution for MBPO but also a model for how a broader taxonomy of failure modes could be constructed. Such a taxonomy would generalize these types of insights into systematic guidance for designing algorithms that are robust across diverse environments and would help practitioners anticipate in advance whether the structure of their MDP is amenable to a particular algorithm choice.

**Usage of Large Language Models (LLMs)**   LLMs were used for paper writing assistance and to aid in brainstorming, software development, and experiment design.

**Reproducibility Statement**    We have taken several steps to ensure the reproducibility of our results. Implementation details for SAC, MBPO, and FTFL are provided in Appendix A, with complete hyperparameter tables included in the appendix. Additional raw return curves, remediation ablations, and environment-specific analyses are reported in Appendix B, Appendix C, and Appendix D, respectively. Our results build on the efficient MBPO reimplementation from Barkley & Fridovich-Keil (2025), which is publicly available.

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

## A SAC AND MBPO IMPLEMENTATION DETAILS

Our FTFL implementation builds directly on the codebase of Barkley & Fridovich-Keil (2025) (github.com/CLeARoboticsLab/STFL), using their default hyperparameters as reproduced below. In both FTFL and MBPO, input normalization is recomputed before every ensemble retraining by calculating independent means and standard deviations for the actions and states in the replay buffer. These statistics are stored and used to unit-normalize transitions sampled for training batches. Target normalization is performed in the same way, independently for rewards and next states at the same cadence, and is additionally required to denormalize model outputs back into the original next-state and reward scales.

Table 1: Hyperparameters used for SAC.

| Hyperparameters | Value |
|---|---|
| Discount ($\gamma$) | 0.99 |
| Warmup steps | 10000 |
| Minibatch size | 256 |
| Optimizer | Adam |
| Learning rate ($\alpha$) | 0.0003 |
| Optimizer $\beta_1$ | 0.9 |
| Optimizer $\beta_2$ | 0.999 |
| Optimizer $\epsilon$ | 0.00015 |
| Networks activation | ReLU |
| Number of hidden layers | 2 |
| Hidden units per layer | 256 |
| Initial temperature ($\alpha_0$) | 1 |
| Replay buffer size | $10^6$ |
| Updates per step | 20 |
| Target network update period | 1 |
| Soft update rate ($\tau$) | 0.995 |

Table 2: Hyperparameters used for MBPO.

| Hyperparameters | Value |
|---|---|
| Ensemble retrain interval | 250 |
| Minibatch size | 256 |
| Optimizer | Adam |
| Ensemble learning rate | 0.0003 |
| Optimizer $\beta_1$ | 0.9 |
| Optimizer $\beta_2$ | 0.999 |
| Optimizer $\epsilon$ | 0.00015 |
| Networks activation | Swish |
| Synthetic ratio | 0.95 |
| Model rollouts per environment step | 400 |
| Number of ensemble layers | 2 |
| Hidden units per layer | 200 |
| Number of elite models | 5 |
| Number of models in ensemble | 7 |
| Model horizon | 1 |

## B  RAW RETURN CURVES FOR SAC, MBPO, FTFL, AND TUNED FTFL IN GYM AND DMC

In this section we provide raw return curves for all seven DMC tasks where MBPO previously failed and for six Gym tasks where MBPO previously succeeded. We compare MBPO to its base off-policy method (SAC), as well as to FTFL and Tuned FTFL. Tuning the hidden units per layer of the model ensemble to 400 only improved performance on the `humanoid` tasks in DMC. For all other tasks, we therefore report FTFL results with the base capacity of 200 units per layer.

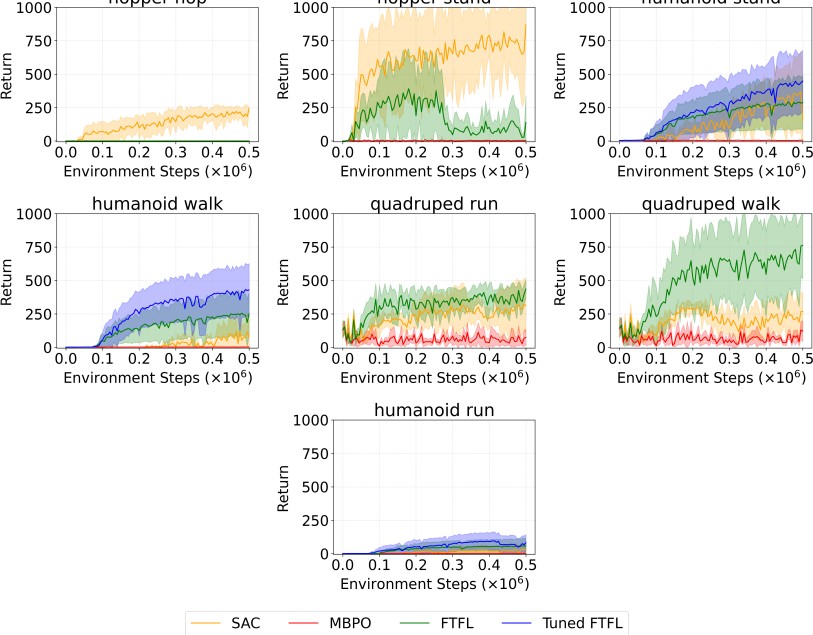

Figure 6: Raw return performance on seven DMC tasks across six seeds each. Solid lines indicate mean and shaded regions denote the standard deviation. Results are shown for FTFL with the original model and for Tuned FTFL with increased model capacity (effective only on `humanoid` tasks).

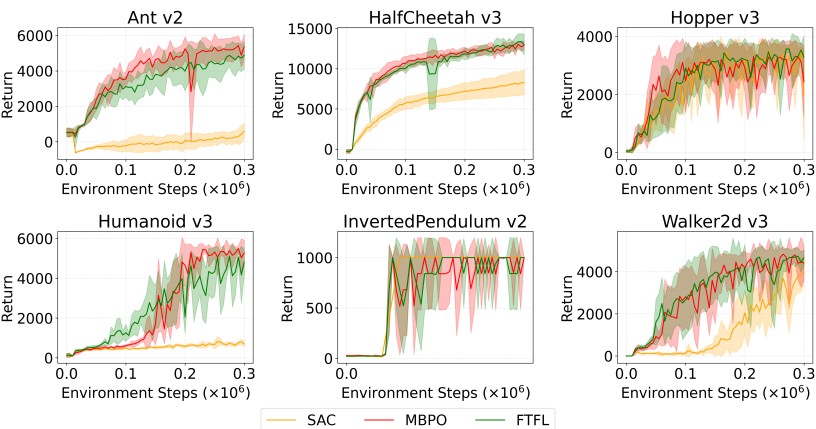

Figure 7: Raw return performance on six gym tasks across six seeds each. Solid lines indicate mean and shaded regions denote the standard deviation.

## C  REMEDIATION ABLATION

In the main text we reported that both target normalization and direct next-state prediction are required for stable policy improvement. To provide full context, we include here the raw return curves for the remediation ablation across all seven DMC tasks where MBPO previously failed. These curves show every combination of residual vs. direct prediction and target normalization vs. no target normalization, complementing the analysis in the main text.

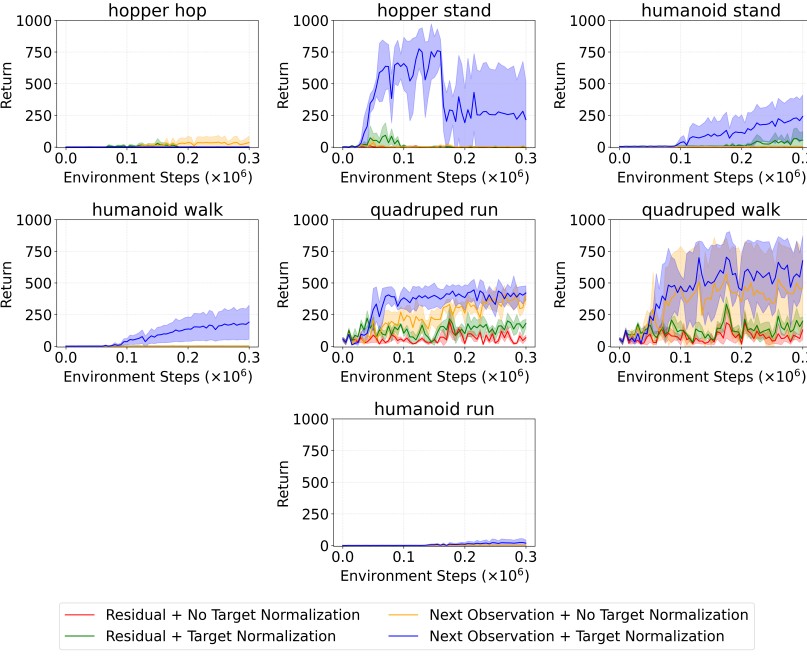

Figure 8: Raw return performance ablation on seven DMC tasks, shown for all combinations of residual vs. next-observation prediction and target normalization vs. no target normalization. Curves are averaged over three seeds per setting, with solid lines indicating means and shaded regions denoting standard deviations.

## D  FTFL AND SAC PERFORMANCE IN HOPPER ENVIRONMENTS WITHOUT TOUCH SENSOR OBSERVATIONS

To further probe the modality-specific challenges in Hopper, we include here the raw return curves for the no-contact ablation. This experiment removes the discontinuous contact-force signal from the `hopper` observation, allowing us to isolate its impact on FTFL's performance. The curves show that eliminating this signal enables FTFL to match SAC on `hopper-stand` but not on `hopper-hop`, complementing the discussion in the main text.

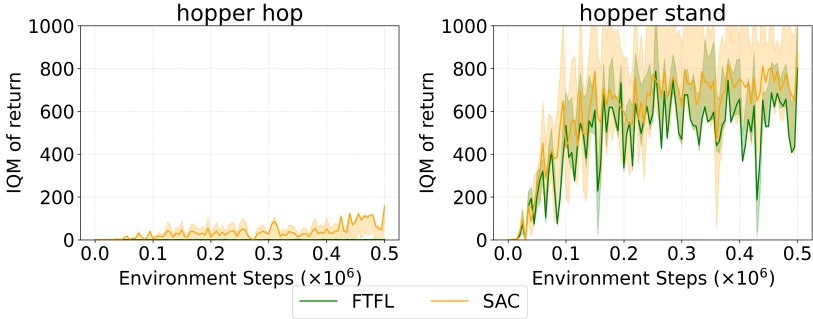

Figure 9: IQM return performance on the `hopper` tasks across six seeds each. Solid lines show interquartile mean (IQM) returns aggregated across six seeds, and shaded regions denote 95% bootstrapped confidence intervals.

