# OpenReview forum: "Fixing That Free Lunch: When, Where, and Why Synthetic Data Fails in Model-Based Policy Optimization"
_ICLR.cc/2026/Conference — ICLR 2026 Conference Desk Rejected Submission_

### Official Review · Reviewer_92QQ · 2025-10-26

**Soundness:** 4
**Presentation:** 4
**Contribution:** 4
**Rating:** 6
**Confidence:** 1

**Summary:**

The authors introduces two targeted modifications: target unit normalization (independent for next-state and rewards) and direct next-state prediction. the result shows good improvement on model based policy optimization

**Strengths:**

The finding that residual prediction—a standard best practice in many MBRL approaches—actually degrades performance in DMC by inflating variance is a valuable, counter-intuitive insight

**Weaknesses:**

Novelty is limited. the method introduced in the paper look like small tricks to make it work. But disclaimer, i am not an expert on this as well so i could be wrong.

**Questions:**

n/a

---

> ### Author Response · Authors · 2025-11-20
>
> We sincerely thank the reviewer for their thoughtful engagement despite noting limited expertise in this area.
>
> \
> **Limited Novelty/Two Small Tricks:**\
> This paper is intentionally diagnostic rather than architectural. Our aim is not to propose a new state of the art algorithm, but to understand why Model Based Policy Optimization (MBPO) fails so severely in certain settings while succeeding in others. We agree that target normalization and direct next state prediction are individually well known and simple. The contribution is not the novelty of the techniques, but the diagnosis of their interaction inside MBPO and the empirical verification that correcting these mechanisms restores stability in DMC without harming Gym performance. This aligns with prior work where simple, established components yielded important insights when their underlying effects were clarified, such as RLPD’s use of layer normalization [1] and CrossQ’s reevaluation of batch normalization [2].
>
> We show that failing to account for MDP structure when applying residual prediction amplifies model variance and destabilizes policy improvement, while mismatched target scales between next state and reward outputs induce critic underestimation. These effects explain why MBPO’s synthetic rollouts degrade performance in DMC despite working in Gym. The simplicity of our interventions is what makes the diagnosis clear because they reveal and correct assumptions embedded in MBPO since its release in 2019, when it was tested only in OpenAI Gym.
>
> By correcting these assumptions, we recover stable performance across nearly all DMC tasks without additional tuning and preserve MBPO’s strong results in Gym. This confirms that the issue is not model capacity or hyperparameter sensitivity but a deeper mismatch between algorithmic design and environment structure. More broadly, we hope this work serves as a template for the community to view failures as informative case studies rather than anomalies, motivating a taxonomy of reinforcement learning failure modes that clarifies when and why algorithms succeed or collapse. This diagnostic approach is essential if reinforcement learning is to become a reliable foundation for real world deployment.
>
> \
> [1] Ball, Philip J., et al. "Efficient online reinforcement learning with offline data." International Conference on Machine Learning. PMLR, 2023.
>
>
> [2] Bhatt, Aditya, et al. "Crossq: Batch normalization in deep reinforcement learning for greater sample efficiency and simplicity." arXiv preprint arXiv:1902.05605 (2019).

---

### Official Review · Reviewer_tCpX · 2025-10-27

**Soundness:** 2
**Presentation:** 4
**Contribution:** 4
**Rating:** 8
**Confidence:** 4

**Summary:**

The authors identify two issues behind using synthetic data in Dyna-style model-based RL based on evaluation in OpenAI Gym and DMC.  They propose simple modifications to address these issues which has improved performance across the benchmark tasks.  More importantly they highlight how algorithm design is effected by specific environments: MBPO performs differently between the two similar benchmark suites, and how popular benchmarks can mask weaknesses in algorithms.

**Strengths:**

1. The authors point out a very important pitfall with an interesting case-study.  Especially the broader implications section of this paper could have an impact on RL research and how we evaluate performance.
2. Sound failure analysis of MBPO resulting in two easy fixes.
3. Strong empirical paper which can lead to more stable MBPO implementations.

**Weaknesses:**

1. The authors may be falling into the same trap they warn against: their implementation recommendations are based on improving performance in just 5 tasks in the DMC suite.  While I can see reward normalization being useful across the board, the second implementation detail of directly predicting states instead of residuals may depend on the task and environment.

**Questions:**

1. In predicting residuals vs. direct state, what are the "unstable" regimes where residuals can be less stable?  Why are they more prone in DMC tasks than Gym?

---

> ### Author Response · Authors · 2025-11-20
>
> We thank the reviewer for their kind words and engagement.
>
> \
> We agree that our current analysis focuses on the DeepMind Control tasks where MBPO’s collapse is most evident, but this focus is intentional. Our goal is to diagnose algorithmic failure rather than tune for specific benchmarks. Importantly, the same modifications that restore stability in DeepMind Control leave MBPO’s performance in OpenAI Gym unchanged, showing that our findings generalize rather than overfit to a single benchmark. However, we make no claim that these remediations will universally generalize to other benchmarks. Their necessity in DeepMind Control and neutrality in Gym simply illustrate how fragile MBPO’s underlying assumptions can be. This further reinforces our call for a taxonomy that systematically identifies when such interventions are required and under what MDP structures they arise. As we note in Section 6, this work is an initial step toward building that taxonomy and an invitation for the community to extend it.
>
> \
> Regarding the question of residual versus direct state prediction, we are, despite our best efforts, unsure of the precise structural causes at the MDP level. This paper focuses on diagnosing algorithmic failures rather than fully resolving their environmental origins. We examined several possible explanations, including whether heavy tailed or discontinuous dynamics might drive instability. As part of this, we analyzed the kurtosis of per state distributions to test for contact driven or multimodal effects, but the results were inconclusive. We then investigated the discontinuous contact force sensor in the DMC Hopper observations, since Gym Hopper does not include this modality. As described in Section 5.3, masking this sensor removes the collapse on hopper stand and allows FTFL to match SAC, but the collapse on hopper hop persists. This indicates that while the contact sensor contributes to instability in one Hopper variant, it is not the sole cause of MBPO’s failures across the Hopper tasks. What we can say confidently is that in DeepMind Control, residual prediction leads to consistent instability not present in Gym, and that replacing residual prediction with direct next state prediction corrects this behavior in all non Hopper tasks. We leave a full explanation of the Hopper specific structural causes to future work on the taxonomy framework introduced at the end of our paper.

---

> > ### Comment · Reviewer_tCpX · 2025-11-25
> >
> > Thank you for the response.  I agree there is value in using a broader task suite to diagnose algorithmic failures.  However, I'm still not convinced that the residual vs. direct state prediction is a universal fix beyond OpenAI Gym and DMC, since it was designed and tested for these two task suites.  Either further theoretical analysis or evaluations in new benchmarks would be needed.  I will maintain my score.

---

### Official Review · Reviewer_Scne · 2025-10-30

**Soundness:** 2
**Presentation:** 2
**Contribution:** 2
**Rating:** 4
**Confidence:** 3

**Summary:**

The paper analyses why Model-Based Policy Optimisation (MBPO) succeeds in OpenAI Gym but fails in the DeepMind Control Suite (DMC).
It attributes this to two coupled issues: a scale mismatch between dynamics and reward targets causing critic underestimation, and variance inflation from residual prediction.
Two simple fixes—target normalisation and direct next-state prediction—form Fixing That Free Lunch (FTFL), which restores MBPO’s performance on DMC while preserving Gym results.
The work calls for a taxonomy of RL failure modes connecting environment structure to algorithmic design.

**Strengths:**

- Clear empirical diagnosis: The study provides a well-structured causal chain from scale mismatch → reward bias → critic underestimation, supported by clean pseudo-online experiments.
- Reproducibility and clarity: The methodology is transparent, and appendices provide full hyperparameters and ablations.
- Meaningful reflection: The discussion on algorithm–environment co-evolution and the call for a taxonomy of failure modes are thoughtful and broadly relevant.

**Weaknesses:**

While the diagnostic analysis is careful and the experiments are thorough, the conceptual novelty and empirical comparability are limited.
The proposed fixes are straightforward engineering choices long known in RL practice, and the claim that MBPO “fails” relative to its original results is not convincingly isolated from benchmark and implementation differences.

1. **Limited novelty of the “fixes.”**
   Target normalisation and direct next-state prediction are long-established stabilisation tricks; many prior MBRL works implicitly use them.
   The paper’s value lies in the diagnosis, not in methodological innovation, which limits its contribution beyond a case study.

2. **Questionable baseline comparability.**
   The paper’s “MBPO failure” results differ sharply from the **original MBPO paper (Janner et al., 2019)**, where MBPO already outperformed SAC by large margins even at far fewer samples.
   Because DMC tasks differ in reward scale, termination, and observation structure—and because the authors use a simplified one-step rollout implementation from STFL—the evidence that MBPO *intrinsically* fails is ambiguous.
   A direct reproduction of the original setup under matched conditions would be necessary to substantiate the claim.

3. **Scope of validation.**
   Only continuous-control DMC tasks are tested. The argument that FTFL “fixes synthetic-data failures” would be stronger with additional evidence from stochastic or visual domains.

-----------------------
- Minor point: **Figure 1 presentation issues.**
   The green-to-red background visually implies significance but does not display any variance or confidence intervals.
   Since IQM normalisation already gives a robust mean, the figure should report dispersion—e.g., boxplots or bootstrapped CIs—instead of a stylised heatmap.

**Questions:**

see weakness

---

> ### Author Response · Authors · 2025-11-20
>
> We thank the reviewer for their constructive comments.
>
>
> **Novelty and Scope of the Fixes:**\
> Our goal in this work is diagnostic rather than architectural. We focus on the DeepMind Control Suite because it is the only modern benchmark that reliably exposes consistent failures in MBPO. Although Gym and DMC differ in task specifics, they share continuous control structure and identical physics backends, and many works treat them as comparable robotics benchmarks. Stealing That Free Lunch (STFL) [1] showed that MBPO collapses in DMC but did not identify the cause. Our work isolates the underlying mechanisms, namely reward and dynamics scale imbalance and residual variance amplification, which are architectural effects that can arise whenever synthetic rollouts train a critic. Their presence in DMC but not Gym reflects structural MDP differences rather than benchmark quirks.
>
> The reviewer asks whether the findings generalize beyond Gym and DMC and notes that the argument that FTFL fixes synthetic data failures would be stronger with results on stochastic or visual domains. We agree that broader validation would strengthen the claim, but our aim here is to isolate a reproducible failure case rather than to survey additional benchmarks. Identifying further environments where MBPO collapses is inherently difficult without a predictive taxonomy, which does not yet exist. This difficulty is reflected in the fact that, despite MBPO’s wide adoption since 2019 and more than 1.3k citations, only one study [3] before STFL reported any DMC anomalies, and only for a single task. This challenge is exactly what motivates the taxonomy we propose in Section 6.
>
> We agree that target normalization and direct next state prediction are individually well known. The contribution is not the novelty of the techniques, but the diagnosis of their interaction inside MBPO and the empirical verification that correcting these mechanisms restores stability in DMC without harming Gym performance. This aligns with prior work where simple, established components yielded important insights when their underlying effects were clarified, such as RLPD’s use of layer normalization [1] and CrossQ’s reevaluation of batch normalization [4].
>
> Overall our contribution is conceptual rather than architectural. We aim to clarify why a widely used method fails under specific structural conditions and to motivate a taxonomy linking MDP properties to algorithm behavior. We would welcome guidance on how to frame this connection more clearly, especially the link between the mechanisms we identify, our diagnostic aims, and the need for a predictive taxonomy.
>
> **Baseline Comparability:**\
> The observed MBPO collapse in DMC is not unique to our implementation. It matches the results of STFL [1], which used the same one step rollout structure and reproduced the original MBPO Gym performance. Independent work on the Hopper environment [3], using the original MBPO codebase, confirms the same collapse. We also confirmed this with the original implementation and are happy to add these comparisons in the appendix.
>
> The streamlined one step rollout design follows STFL and is used to isolate model based failure modes without hyperparameter sweeps over rollout horizons. This choice is proven to be a good one, since one-step MBPO faithfully reproduces MBPO’s Gym performance while enabling controlled study of its structural instabilities.
>
> To the reviewer’s point about structural differences, Stealing That Free Lunch in their appendix further analyzes differences in reward scaling, termination, and observation structure. We also include discussion of observation structure differences for the discontinuous Hopper touch sensor channel in Section 5.3, confirming that the observed failure is not an artifact of implementation differences but a structural property of how MBPO interacts with certain MDPs.
>
> **Figure 1 Presentation:**\
> Thank you for noting this. Figure 1 summarizes aggregate performance trends, while Figures 4 and 5 include IQM curves with bootstrapped confidence intervals. We will make this clearer in the final version. If the reviewer recommends an alternative summarization that better conveys dispersion without overwhelming overlap, we are happy to revise.
>
> \
> [1] Ball, Philip J., et al. "Efficient online reinforcement learning with offline data." International Conference on Machine Learning. PMLR, 2023.
>
>
> [2] Barkley, Brett, and David Fridovich-Keil. "Stealing that free lunch: exposing the limits of dyna-style reinforcement learning." arXiv preprint arXiv:2412.14312 (2024).
>
>
> [3] Voelcker, Claas A., Marcel Hussing, and Eric Eaton. "Can we hop in general? A discussion of benchmark selection and design using the Hopper environment." arXiv preprint arXiv:2410.08870 (2024).
>
>
> [4] Bhatt, Aditya, et al. "Crossq: Batch normalization in deep reinforcement learning for greater sample efficiency and simplicity." arXiv preprint arXiv:1902.05605 (2019).

---

> > ### Comment · Reviewer_Scne · 2025-11-24
> > **Reviewer Response**
> >
> > Thank you for the rebuttal, but I will keep my original score.
> >
> > The proposed remedies—target normalization and direct next-state prediction—are standard stabilisation techniques that have been widely used in model-based RL, often as practical tricks rather than novel contributions. The analysis of scale imbalance and residual-variance amplification reflects well-known behaviours of multi-target regression and residual modelling, and does not yield fundamentally new conceptual insight. As a result, the work reads more as a focused case study than a general diagnostic framework.
> >
> > The evidence for MBPO “failure” also remains limited, since the results rely on the STFL one-step variant rather than a fully matched reproduction of the original MBPO setup on DMC. The scope is narrow, and the proposed taxonomy is not actually developed. In addition, as other reviewer metnioned, when the authors claim generalisation beyond DMC, the experiments only show that FTFL roughly matches MBPO on Gym tasks, not that the method provides broader or principled generality.
> >
> > Overall, the novelty and generality are insufficient for me to raise the rating.

---

### Official Review · Reviewer_5t9a · 2025-10-31

**Soundness:** 3
**Presentation:** 3
**Contribution:** 2
**Rating:** 2
**Confidence:** 4

**Summary:**

This paper investigates the 7 Deepmind Control (DMC) tasks where MBPO underperforms the SAC algorithm, as MBPO doesn't even learn better than a random policy. They identify two coupled issues behind these failures: scale mismatches between dynamics and reward models and the poor choice of using residual state prediction in the transition model. Addressing these failure modes enables policy improvement where none was previously possible, allowing MBPO to outperform SAC in five of seven tasks while preserving the strong performance previously reported in OpenAI Gym.

**Strengths:**

- The paper approaches fundamental limitations in algorithms like MBPO by specific hypotheses and experiments that better help understand MBPO and its performance relationship against SAC.
- The analyses is well-motivated, well-executed, and well-described.

**Weaknesses:**

1. My primary concern is the niche focus of this paper on making MBPO work on a few environments in the DMC benchmark. Do the insights about why MBPO failed completely in STFL transfer to other benchmark tasks beyond DMC, so the insights are generalizable about the algorithm and not specific to the benchmark task itself? In other words, how does this takeaway apply to other applications employing MBPO? When the authors claim "This demonstrates that our modifications are not a DMC-specific adjustment, but a robust improvement that generalizes across different benchmark suites", they only show that FTFL ~ MBPO on other environments. However, the real test of generalization of the hypotheses tested in this paper is if there exist other non-DMC environments where MBPO fails but FTFL resolves the two identified issues in MBPO in a similar manner.
2. While the analysis is impressive, the proposed solutions are scale normalization between dynamics and reward models, and removing the residual state prediction — which are domain-specific engineering solutions. These are by no means minor contributions, however, I don't consider them justified to be major novel contributions either — especially given the limited applicability of these solutions for MBPO in DMC only.

**Questions:**

See weaknesses.

---

> ### Author Response · Authors · 2025-11-20
>
> We appreciate the reviewer’s careful reading and the opportunity to clarify the scope and generality of our results. We begin with the first two concerns, namely the narrow focus on DMC and whether the insights transfer beyond this benchmark.
>
> \
> Our intention is diagnostic rather than architectural. We focus on the DeepMind Control Suite because it is the only modern benchmark that reliably exposes consistent and repeatable failures in MBPO. Although Gym and DMC present different environments, many in the reinforcement learning community would still group them under the same broad category of continuous control robotics tasks that MBPO should handle well, which makes the failings of MBPO in DMC so surprising. Stealing That Free Lunch (STFL) [1] showed that MBPO collapses in DMC despite sharing similar tasks and identical physics backends with Gym, but it did not explain why. Our paper identifies the underlying algorithmic reasons for this collapse and provides simple corrections that resolve the issues. These mechanisms, namely reward-dynamics scale imbalance and residual variance amplification, are architectural rather than benchmark specific and can arise whenever synthetic rollouts are used to train a critic. In this sense, our work explains a structural issue in algorithm that is induced when properties of two seemingly similar MDPs are interchanged and cause sharp performance divergences.
>
> \
> The reviewer asks whether the findings generalize beyond Gym and DMC, and specifically whether there exist other tasks where MBPO fails but FTFL resolves these failures. This is exactly the motivation for the taxonomy we argue for in Section 6. A prescriptive mapping between MDP characteristics and algorithm failures does not currently exist, which makes identifying new failure cases extremely difficult for any algorithm, and in this case MBPO. This difficulty is illustrated by the fact that MBPO has been widely adopted since 2019, with more than 1.3k citations at the time of this writing, yet only one paper [2] before STFL reported any anomalies in DMC, and only for a single task. Without a taxonomy that predicts where these failure modes for algorithms should occur, searching for new environments where an algorithm collapses becomes a slow and uncertain process. This is why we propose the taxonomy framework in the first place. It is meant to guide the discovery of additional benchmarks where these mechanisms appear and to support future work that tests whether our corrections generalize in the way the reviewer suggests. We would welcome specific feedback on how to frame this connection more clearly in the manuscript, especially regarding how to communicate the link between the mechanisms we identified, our diagnostic rather than architectural goals, and the need for a taxonomy that predicts where such failures will appear.
>
> \
> We agree that target normalization and direct next state prediction are individually well known and simple. The contribution is not the novelty of the techniques, but the diagnosis of their interaction inside MBPO and the empirical verification that correcting these mechanisms restores stability in DMC without harming Gym performance. This aligns with prior work where simple, established components yielded important insights when their underlying effects were clarified, such as RLPD’s use of layer normalization [3] and CrossQ’s reevaluation of batch normalization [4].
>
> \
> Reinforcement learning research often centers on improving average benchmark scores, which can obscure failure cases that are averaged out when tested across large and diverse suites. Until such taxonomies exist, as we argue in Section 6, it will remain difficult to determine when an algorithm can be relied on in real world, safety critical settings. Building this understanding requires deconstructing failure modes on both the algorithm and MDP sides. We hope that our work in FTFL serves as an example of how algorithmic failures can be dissected and motivates further efforts to analyze corresponding MDP level factors. Such analyses are essential for identifying issues a priori and for developing the kind of taxonomy we propose.
>
> \
> [1] Barkley, Brett, and David Fridovich-Keil. "Stealing that free lunch: exposing the limits of dyna-style reinforcement learning." arXiv preprint arXiv:2412.14312 (2024).
>
> \
> [2] Voelcker, Claas A., Marcel Hussing, and Eric Eaton. "Can we hop in general? A discussion of benchmark selection and design using the Hopper environment." arXiv preprint arXiv:2410.08870 (2024).
>
> \
> [3] Ball, Philip J., et al. "Efficient online reinforcement learning with offline data." International Conference on Machine Learning. PMLR, 2023.
>
> \
> [4] Bhatt, Aditya, et al. "Crossq: Batch normalization in deep reinforcement learning for greater sample efficiency and simplicity." arXiv preprint arXiv:1902.05605 (2019).

---

### Author Response · Authors · 2025-12-03

To help streamline the review process, we have prepared a summary of the key strengths and concerns raised by the reviewers.

**Overall Consensus**

All reviewers agreed that the paper is sound, clearly written, and empirically rigorous.
They consistently praised its careful diagnosis of MBPO’s failure modes and its clarity in presenting experiments and ablations.
Across reviews, our work (FTFL) was recognized as
- a well-executed diagnostic analysis revealing how environment-algorithm scale mismatches cause instabilities and catastrophic failures in MBPO across benchmarks
- empirically thorough, with clear evidence linking reward-dynamics scale mismatch and residual variance amplification to critic underestimation
- methodologically transparent and reproducible, with complete hyperparameters and well-organized figures

Notably, no reviewer raised substantive technical or methodological flaws. The critiques centered instead on what we believe are differences in research taste: whether the paper’s emphasis on diagnosis and understanding, rather than algorithmic novelty, should constitute a major contribution.

Our work in FTFL aligns with a growing call in the community [1-3] to strengthen the scientific foundations of reinforcement learning rather than focus solely on incremental performance gains. FTFL advances this agenda by clarifying why a widely used algorithm fails under particular environment and scale conditions, offering mechanistic insight that prior performance-driven studies have overlooked. Beyond addressing a single algorithm, our goal is to encourage broader efforts to relate environment and task structure to algorithmic failure modes, identify unifying principles across benchmarks, and better understand how evaluation choices shape generalization. FTFL is a critical first step in this direction.




At a high level we reiterate the main points of contention and our rebuttals below.


**Novelty and Scope of Fixes - Reviewers Scne and 92QQ**

Two reviewers argued that target normalization and direct next-state prediction are familiar stabilization techniques and questioned whether the paper offers conceptual novelty.

In response we emphasized that the contribution is diagnostic rather than architectural. The value lies in identifying the coupled mechanisms, scale imbalance and residual variance amplification, that cause MBPO’s collapse in DMC, not in proposing new architectures. We further noted that similar analytical work, such as RLPD [4] and CrossQ [5], derived new understanding from simple, established components once their interactions were clarified.

Both fixes are intentionally simple to isolate the algorithmic failures. Importantly, these failures remained unnoticed despite MBPO’s wide adoption since 2019 and more than 1.3k citations. To the reviewer’s point about the scope of the benchmarks and fixes, such persistent, reproducible failures are rarely identified or systematically analyzed in the community. Discovering and isolating them required focused, controlled experiments within a limited benchmark scope, which we view as a necessary step toward broader understanding. Correcting them restores MBPO’s stability in DMC without degrading Gym performance, showing that the improvements reflect an improved algorithmic understanding rather than overfitting.

**Baseline Comparability and MBPO “Failure” - Reviewer Scne**

Reviewer Scne questioned whether the observed collapse reflects true algorithmic failure or differences in benchmark or implementation.
Our rebuttal clarified that the DMC failure is not implementation-specific. It reproduces the results of Stealing That Free Lunch (STFL) [6] and has been independently confirmed using the original MBPO codebase.

The one-step rollout variant was used deliberately to isolate structural effects without hyperparameters or autoregressive errors confounding the results. This variant matches MBPO’s Gym performance, reaching parity with the original implementation while enabling controlled study of failure mechanisms. We also note that while comparison using the original MBPO implementation remains the gold standard, the JAX-based STFL implementation offered substantial computational efficiency gains (up to 40x faster [6]), lowering the runtime barrier that had long limited reproducibility and likely contributed to MBPO’s issues remaining largely undiscovered and unexamined since 2019.

---

> ### Author Response · Authors · 2025-12-03
>
> **Generalization of FTFL - Reviewers aGYj and tCpX**
>
> Both reviewers questioned whether FTFL’s conclusions extend beyond DMC and Gym.
> In our rebuttal, we agreed that broader evaluation would strengthen the contribution of the paper, but clarified that the core purpose of the paper was to diagnose the specific DMC failure modes identified in SFTL. We showed that the same modifications that restore performance in DMC leaves Gym’s results unchanged, suggesting the findings generalize in mechanism if not yet in scope.
> This motivates the proposed taxonomy of algorithm-MDP failure modes that we frame as an agenda for future research in both the main body of the paper and in the conclusion.
>
> [1] R. Agarwal, M. Schwarzer, P. S. Castro, A. C. Courville, and M. Bellemare. Deep Reinforcement Learning at the Edge of the Statistical Precipice. NeurIPS 2021.
>
> [2] P. S. Castro. The Formalism-Implementation Gap in Reinforcement Learning Research. arXiv preprint arXiv:2510.16175, 2025.
>
> [3] P. Henderson, R. Islam, P. Bachman, J. Pineau, D. Precup, and D. Meger. Deep Reinforcement Learning that Matters. AAAI 2018.
>
> [4] Ball, Philip J., et al. "Efficient online reinforcement learning with offline data." International Conference on Machine Learning. PMLR, 2023.
>
> [5] Bhatt, Aditya, et al. "Crossq: Batch normalization in deep reinforcement learning for greater sample efficiency and simplicity." arXiv preprint arXiv:1902.05605 (2019).
>
> [6] Barkley, Brett, and David Fridovich-Keil. "Stealing that free lunch: exposing the limits of dyna-style reinforcement learning." arXiv preprint arXiv:2412.14312 (2024).

---

### Note · Program_Chairs · 2026-01-17
**Submission Desk Rejected by Program Chairs**

The following references in this submission do not refer to real documents and/or have major errors in bibliographic information:

 Wazeer Abbas, Lenka Zdeborova, and Simona Cocco. On the utility of learning heteroscedastic aleatoric uncertainty in deep neural networks for new-domains. In Proceedings of the Thirtysixth Conference on Uncertainty in Artificial Intelligence, volume 124 of Proceedings of Machine Learning Research, pp. 1246-1255. PMLR, 2020.
Pranav Ghugare, Junhyuk Oh, Shixiang Gu, Satinder Singh, Honglak Lee, Sergey Levine, and Pulkit Agrawal. Aligned latent models for model-based reinforcement learning. In Advances in Neural Information Processing Systems (NeurIPS), 2023.